# BEYOND PRIORITIZED REPLAY: SAMPLING STATES IN MODEL-BASED RL VIA SIMULATED PRIORITIES

## ABSTRACT

The prioritized Experience Replay (ER) method has attracted great attention; however, there is little theoretical understanding of such prioritization strategy and why they help. In this work, we revisit prioritized ER and, in an ideal setting, show equivalence to minimizing cubic loss, providing theoretical insight into why it improves upon uniform sampling. This theoretical equivalence highlights two limitations of current prioritized experience replay methods: insufficient coverage of the sample space and outdated priorities of training samples. This motivates our model-based approach, which does not suffer from these limitations. Our key idea is to actively search for high priority states using gradient ascent. Under certain conditions, we prove that the hypothetical experiences generated from these states are sampled proportionally to approximately true priorities. We also characterize the distance between the sampling distribution of our method and the true prioritized sampling distribution. Our experiments on both benchmark and application-oriented domains show that our approach achieves superior performance over baselines.

## 1 INTRODUCTION

Using hypothetical experience simulated from an environment model can significantly improve sample efficiency of RL agents (Ha & Schmidhuber, 2018; Holland et al., 2018; Pan et al., 2018; Janner et al., 2019; van Hasselt et al., 2019). Dyna (Sutton, 1991) is a classical MBRL architecture where the agent uses real experience to updates its policy as well as its reward and dynamics models. In-between taking actions, the agent can get hypothetical experience from the model to further improve the policy.

An important question for effective Dyna-style planning is *search-control*: from what states should the agent simulate hypothetical transitions? On each planning step in Dyna, the agent has to select a state and action from which to query the model for the next state and reward. This question, in fact, already arises in what is arguably the simplest variant of Dyna: Experience Replay (ER) (Lin, 1992). In ER, visited transitions are stored in a buffer and at each time step, a mini-batch of experiences is sampled to update the value function. ER can be seen as an instance of Dyna, using a (limited) non-parametric model given by the buffer (see van Seijen & Sutton (2015) for a deeper discussion). Performance can be significantly improved by sampling proportionally to priorities based on errors, as in prioritized ER (Schaul et al., 2016; de Bruin et al., 2018), as well as specialized sampling for the off-policy setting (Schlegel et al., 2019).

Search-control strategies in Dyna similarly often rely on using priorities, though they can be more flexible in leveraging the model rather than being limited to only retrieving visited experiences. For example, a model enables the agent to sweep backwards by generating predecessors, as in prioritized sweeping (Moore & Atkeson, 1993; Sutton et al., 2008; Pan et al., 2018; Corneil et al., 2018). Other methods have tried alternatives to error-based prioritization, such as searching for states with high reward (Goyal et al., 2019), high value (Pan et al., 2019) or states that are difficult to learn (Pan et al., 2020). Another strategy is to directly generate hypothetical experiences from trajectory optimization algorithms (Gu et al., 2016). These methods are all supported by nice intuition, but as yet lack solid theoretical reasons for why they can improve sample efficiency.

In this work, we provide new insights about how to choose the sampling distribution over states from which we generate hypothetical experience. In particular, we theoretically motivate why error-based prioritization is effective, and provide a mechanism to generate states according to more

accurate error estimates. We first prove that $l_2$ regression with error-based prioritized sampling is equivalent to minimizing a cubic objective with uniform sampling in an ideal setting. We then show that minimizing the cubic power objective has a faster convergence rate during early learning stage, providing theoretical motivation for error-based prioritization. The theoretical understanding illuminates two issues of prioritized ER: insufficient sample space coverage and outdated priorities. To overcome the limitations, we propose a search-control strategy in Dyna that leverages a model to simulate errors and to find states with high expected error. Finally, we demonstrate the efficacy of our method on various benchmark domains and an autonomous driving application.

## 2 PROBLEM FORMULATION

We formalize the problem as a Markov Decision Process (MDP), a tuple $(\mathcal{S}, \mathcal{A}, \mathbb{P}, R, \gamma)$ including state space $\mathcal{S}$, action space $\mathcal{A}$, probability transition kernel $\mathbb{P}$, reward function $R$, and discount rate $\gamma \in [0, 1]$. At each environment time step $t$, an RL agent observes a state $s_t \in \mathcal{S}$, and takes an action $a_t \in \mathcal{A}$. The environment transitions to the next state $s_{t+1} \sim \mathbb{P}(\cdot|s_t, a_t)$, and emits a scalar reward signal $r_{t+1} = R(s_t, a_t, s_{t+1})$. A policy is a mapping $\pi : \mathcal{S} \times \mathcal{A} \to [0, 1]$ that determines the probability of choosing an action at a given state.

The agent's objective is to find an optimal policy. A popular algorithm is Q-learning (Watkins & Dayan, 1992), where parameterized action-values $Q_\theta$ are updated using $\theta = \theta + \alpha \delta_t \nabla_\theta Q_\theta(s_t, a_t)$ for step-size $\alpha > 0$ with TD-error $\delta_t \stackrel{\text{def}}{=} r_{t+1} + \gamma \max_{a' \in \mathcal{A}} Q_\theta(s_{t+1}, a') - Q_\theta(s_t, a_t)$. The policy is defined by acting greedily w.r.t. these action-values. ER is critical when using neural networks to estimate $Q_\theta$, as used in DQN (Mnih et al., 2015), both to stabilize and speed up learning. MBRL has the potential to provide even further sample efficiency improvements.

We build on the Dyna formalism (Sutton, 1991) for MBRL, and more specifically the recently proposed HC-Dyna (Pan et al., 2019)

---

**Algorithm 1** HC-Dyna: Generic framework

**Input:** hill climbing crit. $h : \mathcal{S} \mapsto \mathbb{R}$, batch-size $b$
Initialize empty search-control queue $B_{sc}$ ; empty
ER buffer $B_{er}$; initialize policy and model $\mathcal{P}$
**for** $t = 1, 2, \ldots$ **do**
    Add $(s_t, a_t, s_{t+1}, r_{t+1})$ to $B_{er}$
    **while** within some budget time steps **do**
        $s \leftarrow s + \alpha_a \nabla_s h(s)$ //hill climbing
        Add $s$ into $B_{sc}$
    **for** $n$ times **do**
        $B \leftarrow \emptyset$
        **for** $b/2$ times **do**
            Sample $s \sim B_{sc}$, on-policy action $a$
            Sample $s', r \sim \mathcal{P}(s, a)$
            $B \leftarrow (s, a, s', r)$
        Sample $b/2$ experiences from $B_{er}$, add to $B$
        Update policy on the mixed mini-batch $B$

---

as shown in Algorithm 1. HC-Dyna provides a particular approach to search-control—the mechanism of generating states or state-action pairs from which to query the model to get next states and rewards (i.e. hypothetical experiences). It is characterized the fact that it generates states by hill climbing on some criterion function $h(\cdot)$. The term Hill Climbing (HC) is used for generality as the vanilla gradient ascent procedure is modified to resolve certain challenges (Pan et al., 2019). Two particular choices have been proposed for $h(\cdot)$: the value function $v(s)$ from Pan et al. (2019) and the gradient magnitude $||\nabla_s v(s)||$ from Pan et al. (2020). The former is used as measure of the utility of visiting a state and the latter is considered as a measure of value approximation difficulty. The hypothetical experience is obtained by first selecting a state $s$, then typically selecting the action $a$ according to the current policy, and then querying the model to get next state $s'$ and reward $r$.

These hypothetical transitions are treated just like real transitions. For this reason, HC-Dyna combines both real experience and hypothetical experience into mini-batch updates. These $n$ updates, performed before taking the next action, are called planning updates, as they improve the action-value estimates— and so the policy—using a model.

However, it should be noted that there are several limitations to the two previous works. First, the HC method proposed by Pan et al. (2019) is mostly supported by intuitions, without any theoretical justification to use the stochastic gradient ascent trajectories for search-control. Second, the HC on gradient norm and Hessian norm of the learned value function Pan et al. (2020) is supported by some suggestive theoretical evidence, but it suffers from great computation cost and zero gradient due to the high order differentiation (i.e., $\nabla_s ||\nabla_s v(s)||$) as suggested by the authors. This paper will introduce our novel HC search-control method motivated by overcoming the limitations of the

prioritized ER method, which has stronger theoretical support than the work by Pan et al. (2019) and improved computational cost comparing with the existed work by Pan et al. (2020).

## 3 A DEEPER LOOK AT ERROR-BASED PRIORITIZED SAMPLING

In this section, we provide theoretical motivation for error-based prioritized sampling. We show that prioritized sampling can be reformulated as optimizing a cubic power objective with uniform sampling. We prove that optimizing the cubic objective provides a faster convergence rate during early learning. Based on these results, we highlight that prioritized ER has two limitations 1) outdated priorities and 2) insufficient coverage of the sample space. This motivates our method in the next section to address the two limitations.

### 3.1 PRIORITIZED SAMPLING AS A CUBIC OBJECTIVE

In the $l_2$ regression, we minimize the mean squared error $\min_\theta \frac{1}{2n} \sum_{i=1}^n (f_\theta(x_i) - y_i)^2$, for training set $\mathcal{T} = \{(x_i, y_i)\}_{i=1}^n$ and function approximator $f_\theta$, such as a neural network. In error-based prioritized sampling, we define the priority of a sample $(x, y) \in \mathcal{T}$ as $|f_\theta(x) - y|$; the probability of drawing a sample $(x, y) \in \mathcal{T}$ is typically $q(x, y; \theta) \propto |f_\theta(x) - y|$. We employ the following form to compute the probabilities:

$$q(x, y; \theta) \overset{\text{def}}{=} \frac{|f_\theta(x) - y|}{\sum_{i=1}^n |f_\theta(x_i) - y_i|} \tag{1}$$

We can show an equivalence between the gradients of the squared objective with this prioritization and the cubic power objective $\frac{1}{3n} \sum_{i=1}^n |f_\theta(x_i) - y_i|^3$. See Appendix A.3 for the proof.

**Theorem 1.** *For a constant $c$ determined by $\theta, \mathcal{T}$, we have*

$$\mathbb{E}_{(x,y)\sim uniform(\mathcal{T})}[\nabla_\theta (1/3)|f_\theta(x) - y|^3] = c\mathbb{E}_{(x,y)\sim q(x,y;\theta)}[\nabla_\theta (1/2)(f_\theta(x) - y)^2]$$

This simple theorem provides an intuitive reason for why prioritized sampling can help improve sample efficiency: the gradient direction of cubic function is sharper than that of the square function when the error is relatively large (Figure 1). Theorem 2 further characterizes the difference between the convergence rates by optimizing the mean square error and the cubic power objective, providing a solid motivation for using error-based prioritized sampling.

**Theorem 2** (Fast early learning). *Consider the following two objectives: $\ell_2(x, y) \overset{\text{def}}{=} \frac{1}{2}(x - y)^2$, and $\ell_3(x, y) \overset{\text{def}}{=} \frac{1}{3}|x - y|^3$. Denote $\delta_t \overset{\text{def}}{=} |x_t - y|$, and $\tilde{\delta}_t \overset{\text{def}}{=} |\tilde{x}_t - y|$. Define the functional gradient flow updates on these two objectives:*

$$\frac{dx_t}{dt} = -\eta \frac{d\{\frac{1}{2}(x_t - y)^2\}}{dx_t}, \frac{d\tilde{x}_t}{dt} = -\eta \frac{d\{\frac{1}{3}|\tilde{x}_t - y|^3\}}{d\tilde{x}_t}. \tag{2}$$

*Given error threshold $\epsilon \geq 0$, define the hitting time $t_\epsilon \overset{\text{def}}{=} \min_t\{t : \delta_t \leq \epsilon\}$ and $\tilde{t}_\epsilon \overset{\text{def}}{=} \min_t\{t : \tilde{\delta}_t \leq \epsilon\}$. For any initial function value $x_0$ s.t. $\delta_0 > 1$, $\exists \epsilon_0 \in (0, 1)$ such that $\forall \epsilon > \epsilon_0, t_\epsilon \geq \tilde{t}_\epsilon$.[1]*

*Proof.* Please see Appendix A.4. Given the same $\epsilon$ and the same initial value of $x$, first we derive $t_\epsilon = \frac{1}{\eta} \cdot \ln\{\frac{\delta_0}{\epsilon}\}, \tilde{t}_\epsilon = \frac{1}{\eta} \cdot (\frac{1}{\epsilon} - \frac{1}{\delta_0})$. Then we analyze the condition on $\epsilon$ to see when $t_\epsilon \geq \tilde{t}_\epsilon$, i.e. minimizing the square error is slower than minimizing the cubic error. $\square$

The above theorem says that when the initial error is relatively large, it is faster to get to a certain low error point with the cubic objective. We can test this in simulation, with the following minimization problems: $\min_{x \geq 0} x^2$ and $\min_{x \geq 0} x^3$. We use the hitting time formulae $t_\epsilon = \frac{1}{\eta} \cdot \ln\{\frac{\delta_0}{\epsilon}\}, \tilde{t}_\epsilon = \frac{1}{\eta} \cdot (\frac{1}{\epsilon} - \frac{1}{\delta_0})$ derived in the proof, to compute the hitting time ratio $\frac{t_\epsilon}{\tilde{t}_\epsilon}$ under different initial values $x_0$ and final error value $\epsilon$. In Figure 1(c)(d), we can see that it usually takes a significantly shorter time for the cubic loss to reach a certain $x_t$ with various $x_0$ values.

---

[1] Finding the exact value of $\epsilon_0$ would require a definition of ordering on complex plane, which leads to $\epsilon_0 = -\frac{1}{W(\log 1/a - 1/a - \pi i)}$ and $W(\cdot)$ is a Wright Omega function, then we have $\tilde{t}_\epsilon \leq t_\epsilon$. Our theorem statement is sufficient for the purpose of characterizing convergence rate.

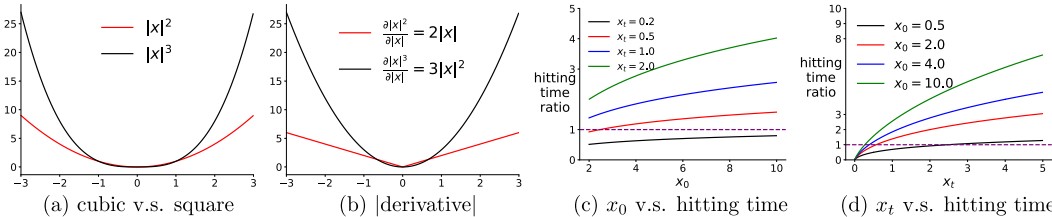

(a) cubic v.s. square    (b) |derivative|    (c) $x_0$ v.s. hitting time    (d) $x_t$ v.s. hitting time

Figure 1: (a) show cubic v.s. square function. (b) shows their absolute derivatives. (c) shows the hitting time ratio v.s. initial value $x_0$ under different target value $x_t$. (d) shows the ratio v.s. the target $x_t$ to reach under different $x_0$. Note that a ratio larger than 1 indicates a longer time to reach the given $x_t$ for the square loss.

**Implications of the above theory.** The equivalence from Theorem 1 inspires us to identify two limitations of the current prioritized ER method: 1) The equivalence requires the priorities of all samples to get updated after the training parameters get updated at each time step. 2) The equivalence requires the prioritized sampling distribution to be calculated on the whole training set; in an online RL setting, at the current time step $t$, we only have visited samples. These visited samples provide a biased training set w.r.t. current policy which likely does not reasonably cover the state space. We will present our approach to overcome the limitations in Section 4. In the next section, we will empirically verify our theoretical findings.

### 3.2 Empirical Demonstration

In this section, we empirically show: 1) the practical performance of the cubic objective; 2) the importance of having sufficient sample space coverage and of updating the priorities of all the training samples; and 3) the reasons for why we should not directly use a high power objective in general. We refer readers to A.6 for missing details and to A.7 for additional experiments.

We conduct experiments on a supervised learning task. We generate a training set $\mathcal{T}, |\mathcal{T}| = 4000$ by uniformly sampling $x \in [-2, 2]$ and adding zero-mean Gaussian noise with standard deviation $\sigma$ to the target $f_{\sin}(x)$ values, where $f_{\sin}(x) = \sin(8\pi x)$ if $x \in [-2, 0)$ and $f_{\sin}(x) = \sin(\pi x)$ if $x \in [0, 2]$. The testing set contains 1k samples and the targets are not noise-contaminated. Pan et al. (2020) show that the high frequency region $[-2, 0]$ is the main source of prediction error. Hence we expect prioritized sampling to make a clear difference in terms of sample efficiency on this dataset.

We compare the following algorithms. **L2**: the $l_2$ regression with uniformly sampling from $\mathcal{T}$. **Full-PrioritizedL2**: the $l_2$ regression with prioritized sampling according to the distribution defined in equation 1, the priorities of *all samples* in the training set are updated after each mini-batch update. **PrioritizedL2**: the only difference with **Full-PrioritizedL2** is that *only* the priorities of those training examples sampled in the mini-batch are updated at each iteration, the rest of the training samples use the original priorities. This resembles the approach taken by vanilla Prioritized ER in the RL setting (Schaul et al., 2016). **Cubic**: minimizing the cubic objective with uniformly sampling. **Power4**: $\min_\theta \frac{1}{n} \sum_{i=1}^n (f_\theta(x_i) - y_i)^4$ with uniformly sampling. We include it to show that there is almost no gain and potential harm by using higher powers.

We use $32 \times 32$ tanh layers for all algorithms and optimize the learning rate from the range $\{0.01, 0.001, 0.0001\}$. Figure 2 (a)-(d) shows the learning curves in terms of testing error of all the above algorithms with various settings.[2] We identify five important observations: 1) with a small mini-batch size 128, there is a significant difference between **Full-PrioritizedL2** and **Cubic**; 2) with increased mini-batch size, although all algorithms perform better, **Cubic** achieves largest improvement and its behavior tends to approximate the prioritized sampling algorithm; 3) as shown in Figure 2 (a), the prioritized sampling does not show advantage when the training set is small; 4) Prioritized $l_2$ without updating all priorities can be significantly worse than vanilla $l_2$ regression (uniform sampling); 5) when increasing the noise standard deviation $\sigma$ from 0.1 to 0.5, all algorithms perform worse and the objectives with *higher power* get *more* hurt.

**The importance of sample space coverage**. Observation 1) and 2) show that a high power objective has to use a much larger mini-batch size to achieve comparable performance with the $l_2$ with

---

[2]We show the testing error as it is the primary concern. The training error has similar comparative performance and is presented in Appendix 3, where we also include additional results with different settings.

Figure 2: Testing RMSE v.s. number of mini-batch updates. (a)(b)(c)(d) show the learning curves with different mini-batch size $b$ or Guassian noises variance $\sigma$ added to the training targets. (a) is using $\sigma = 0.1$ and a **smaller** training set (**solid** line for $|\mathcal{T}| = 800$, **dotted** line for $|\mathcal{T}| = 1600$) than others but has the same testing set size. (e) shows the a corresponding experiment in RL setting on the classical mountain car domain. The results are averaged over 50 random seeds on (a)-(d) and 30 on (e). The shade indicates standard error.

prioritized sampling. A possible reason is that prioritized sampling allows us to immediately get many samples from those high error region. Uniformly sampling, on the other hand, can get fewer of those samples with a limited mini-batch size. This motivates us to test prioritized sampling with a small training set where both algorithms get fewer samples. Figure 2(a) together with (b) indicate that prioritized sampling needs sufficient samples across the sample space to maintain advantage. This requirement is intuitive but it illuminates a serious **limitation of prioritized ER in RL**: only those visited real experiences from the ER buffer can get sampled. If the state space is large, the ER buffer likely contains only a small subset of the state space, indicating a very small training set.

**Thorough priority updating.** Observation 4) highlights the importance of using an up-to-date sampling distribution at each time step. Outdated priorities change the sampling distribution in an unpredictable manner and the learning performance can degrade. We further verify this phenomenon on the classical Mountain Car domain (Sutton & Barto, 2018; Brockman et al., 2016). Figure 2(e) shows the evaluation learning curves of different variants of Deep Q networks (DQN) corresponding to the supervised learning algorithms. We use a small $16 \times 16$ ReLu NN as the $Q$-function. We expect that a small NN should highlight the issue of priority updating: every mini-batch update potentially perturbs the values of many other states. Hence it is likely that many experiences in the ER buffer have the wrong priorities without thorough priority updating. We do indeed find this to be the case, with Full-PrioritizedER performing significantly better.

**Regarding high power objectives.** As we discussed above, observation 1) and 2) tell us that that high power objective likely requires a large mini-batch size. Ideally, it would use a batch algorithm, i.e. the whole training set, for the improved convergence rate to manifest. This requirement makes the algorithm not scalable to larger training dataset. Observation 5) indicates another reason for why a high power objective should not be preferred: it augments the effect of noise added to the target variables. In Figure 2(d), the **Power4** objective suffers most from the increased target noise.

## 4 ADDRESSING THE LIMITATIONS PRIORITIZED REPLAY: ACQUIRING SAMPLES VIA SIMULATED PRIORITIES ON CONTINUOUS DOMAINS

In this section, we propose a method to mitigate the limitations of the conventional prioritized ER method mentioned in the above section. We start by the following theorem. We denote $\mathbb{P}^\pi(s', r|s)$ as the transition probability given a policy $\pi$.

**Theorem 3.** *Sampling method. Given the state $s \in \mathcal{S}$, let $v^\pi(\cdot; \theta) : \mathcal{S} \mapsto \mathbb{R}$ be a differentiable value function under policy $\pi$ parameterized by $\theta$. Define: $y(s) \stackrel{\text{def}}{=} \mathbb{E}_{r,s' \sim \mathcal{P}^\pi(s',r|s)}[r + \gamma v^\pi(s'; \theta)]$, and denote the TD error as $\delta(s, y; \theta_t) \stackrel{\text{def}}{=} y(s) - v(s; \theta_t)$. Given some initial state $s_0 \in \mathcal{S}$, define the state sequence $\{s_i\}$ as the one generated by state updating rule $s_{i+1} \leftarrow s_i + \alpha_a \nabla_s \log |\delta(s_i, y(s_i); \theta_t)| + X_i$, where $\alpha_a$ is a sufficiently small stepsize and $X_i$ is a Gaussian random variable with some constant variance.[3] Then the sequence $\{s_i\}$ converges to the distribution $p(s) \propto |\delta(s, y(s))|$.*

The proof is a direct consequence of the convergent behavior of Langevin dynamics stochastic differential equation (SDE) (Roberts, 1996; Welling & Teh, 2011; Zhang et al., 2017). We include a brief discussion and background knowledge in the Appendix A.2.

---

[3]The stepsize and variance affects the temperature parameter. We avoid introducing too much notation here and simply treat the two as a hyper-parameters in the implementation. We fix one setting across experiments.

In practice, we can compute the state value estimate by $v(s) = \max_a Q(s, a; \theta_t)$ as suggested by Pan et al. (2019). In the case that a true environment model is not available, we have to compute an estimate $\hat{y}(s)$ of $y(s)$ by a learned model. Then at each time step $t$, states approximately following the distribution $p(s) \propto |\delta(s, y(s))|$ can be generated by

$$s \leftarrow s + \alpha_a \nabla_s \log |\hat{y}(s) - \max_a Q(s, a; \theta_t)| + X, \tag{3}$$

where $X$ is a Gaussian random variable with zero-mean and some small variance. In the implementation, observing that $\alpha_a$ is small, we consider $\hat{y}(s)$ as a constant given a state $s$ without backpropagating through it. We provide an upper bound in the below theorem for the difference between the sampling distribution acquired by the true model and the learned model. We denote the transition probability distribution under policy $\pi$ and the true model as $\mathcal{P}^\pi(r, s'|s)$, and the learned model as $\hat{\mathcal{P}}^\pi(r, s'|s)$. Let $p(s)$ and $\hat{p}(s)$ be the convergent distributions described in Theorem 3 by using the true and learned models respectively. Let $d_{tv}(\cdot, \cdot)$ be the total variation distance between the two probability distributions. Define

$$u(s) \overset{\mathsf{def}}{=} |\delta(s, y(s))|, \hat{u}(s) \overset{\mathsf{def}}{=} |\delta(s, \hat{y}(s))|, Z \overset{\mathsf{def}}{=} \int_{s \in \mathcal{S}} u(s) ds, \hat{Z} \overset{\mathsf{def}}{=} \int_{s \in \mathcal{S}} \hat{u}(s) ds.$$

Then we have the following bound. Please see Appendix A.5 for the proof and further interpretations.

**Theorem 4.** *Assume: 1) the reward magnitude is bounded $|r| \le R_{max}$ and define $V_{max} \overset{\mathsf{def}}{=} \frac{R_{max}}{1-\gamma}$; 2) the largest model error for a single state is some small value: $\epsilon_s \overset{\mathsf{def}}{=} \max_s d_{tv}(\mathcal{P}^\pi(\cdot|s), \hat{\mathcal{P}}^\pi(\cdot|s))$ and the total model error is bounded, i.e. $\epsilon \overset{\mathsf{def}}{=} \int_{s \in \mathcal{S}} \epsilon_s ds < \infty$. Then $\forall s \in \mathcal{S}, |p(s) - \hat{p}(s)| \le \min(\frac{V_{max}(p(s)\epsilon + \epsilon_s)}{\hat{Z}}, \frac{V_{max}(\hat{p}(s)\epsilon + \epsilon_s)}{Z})$.*

**Algorithmic details.** We present our algorithm called Dyna-TD (Temporal Difference error) in the Algorithm 3 in Appendix A.6. Our algorithm follows Algorithm 1, particularly, we choose the function $h(s) \overset{\mathsf{def}}{=} \log |\hat{y}(s) - \max_a Q(s, a; \theta_t)|$, i.e. run the updating rule 3 to generate states.

**Empirical verification of sampling distribution.** We validate the efficacy of our sampling method by empirically examining the distance between the sampling distribution acquired by our gradient ascent rule in equation 3 (denoted as $p_1(\cdot)$) and the desired distribution computed by thorough priority updating $p^*(\cdot)$ of all states under the current parameter on the GridWorld domain (Pan et al., 2019) (Figure 3(a)), where the probability density can be conveniently approximated by discretization. We record the distance change when we train our Algorithm 3. The distance between the sampling distribution fo Prioritized ER (denoted as $p_2(\cdot)$) is also included for comparison. All those distributions are computed by normalizing visitation counts on the discretized $50 \times 50$ GridWorld. We compute the distances of $p_1, p_2$ to $p^*$ by two sensible weighting schemes: 1) on-policy weighting: $\sum_{j=1}^{2500} d^\pi(s_j)|p_i(s_j) - p^*(s_j)|, i \in \{1, 2\}$, where $d^\pi$ is approximated by uniformly sample 3k states from a recency buffer; 2) uniform weighting: $\frac{1}{2500} \sum_{j=1}^{2500} |p_i(s_j) - p^*(s_j)|, i \in \{1, 2\}$. All details are in Appendix A.6

Figure 3(b)(c) shows that our algorithm Dyna-TD, either with a true or an online learned model, maintains a significantly closer distance to the desired sampling distribution $p^*$ than PrioritizedER under both weighting schemes. Furthermore, despite the mismatch between implementation and our above Theorem 3—namely that Dyna-TD may not run enough gradient steps to reach stationary distribution—the induced sampling distribution is quite close to the one by running long gradient steps (Dyna-TD-Long), which we expect to reach stationary behavior. This indicates that the we can reduce the time cost by lowering the number of gradient steps, while keep the sampling distribution similar. In Figure 3(d), we further verify that given the same time budget, our algorithm achieves better performance, despite the fact that DQN and PrioritizedER are able to process many more samples. This makes the additional time spent on search-control worth it.

**Sample space coverage.** To further illuminate that our method indeed enables a broader coverage of the model-free ER method, we visualize the DQN's ER state distributions trained with and without prioritization respectively and our algorithm's search-control queue state distribution. Figure 4 shows that there is a significant difference between ER's and our queue's distributions. Specifically, our search-control queue distribution looks more uniformly distributed across the whole state space with slightly higher density in the middle. This concentration corresponds to Figure 5 from the work

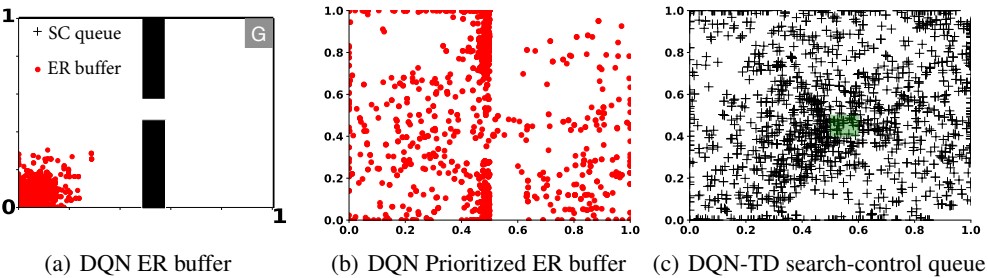

Figure 3: (a) shows the GridWorld taken from Pan et al. (2019). The state space is $\mathcal{S} = [0,1]^2$, and the agent starts from the left bottom and should learn to take action from $\mathcal{A} = \{up, down, right, left\}$ to reach the right top within as few steps as possible. (b) shows the distance change as a function of training steps. The **dashed** line corresponds to our algorithm with an online learned model. The corresponding evaluation learning curve is in the Figure 5(c). (d) shows the policy evaluation performance as a function of running time (seconds). All results are averaged over 20 random seeds and the shade indicates standard error.

by Pan et al. (2020), as the agent must learn to pass the small hole (i.e., a bottleneck region) to get to the goal area. The significantly broader coverage of our search-control queue distribution possibly explains the superior performance of our algorithm in Figure 5 (c)(d).

| (a) DQN ER buffer | (b) DQN Prioritized ER buffer | (c) DQN-TD search-control queue |
|---|---|---|

Figure 4: (a) (b) shows the ER buffer state distributions trained by regular ER and prioritized ER respectively. (c) shows the search-control queue state distribution of our Dyna-TD algorithm after training for the same number of environment time steps. It can be seen that our algorithm has a much broader coverage of the sample space.

## 5 EXPERIMENTS

In this section, we empirically show that our algorithm achieves stable and consistent performance across different settings. We first show the overall comparative performance on various benchmark domains. We then show that our algorithm Dyna-TD is more robust to environment noise than PrioritizedER. Last, we demonstrate the practical utility of our algorithm on an autonomous driving vehicle application. Note that Dyna-TD uses the same hill climbing parameter settings across all benchmark domains. We refer readers to the Appendix A.6 for any missing details.

**Baselines.** We include the following baseline competitors. **ER** is DQN with a regular ER buffer without prioritized sampling. **PrioritizedER** uses a priority queue to store visited experiences and each experience is sampled proportionally to its TD error magnitude. Note that, as per the original paper (Schaul et al., 2016), after each mini-batch update, only the priorities of those samples in the mini-batch are updated. **Dyna-Value** (Pan et al., 2019) is the Dyna variant which performs hill climbing on value function to acquire states to populate the search-control queue. **Dyna-Frequency** (Pan et al., 2020) is the Dyna variant which performs hill climbing on the norm of the gradient of the value function to acquire states to populate the search-control queue.

**Overall Performance.** Figure 5 shows the overall performance of different algorithms on Acrobot, CartPole, GridWorld (Figure 3(a)) and MazeGridWorld (Figure 5(g)). Our key observations are: 1) Dyna-Value or Dyna-Frequency may converge to a sub-optimal policy when using a large number of planning steps; 2) Dyna-Frequency has clearly inconsistent performance across different domains; 3) our algorithm performs the best in most cases: even with an online learned model, our algorithm outperforms others on most of the tasks.

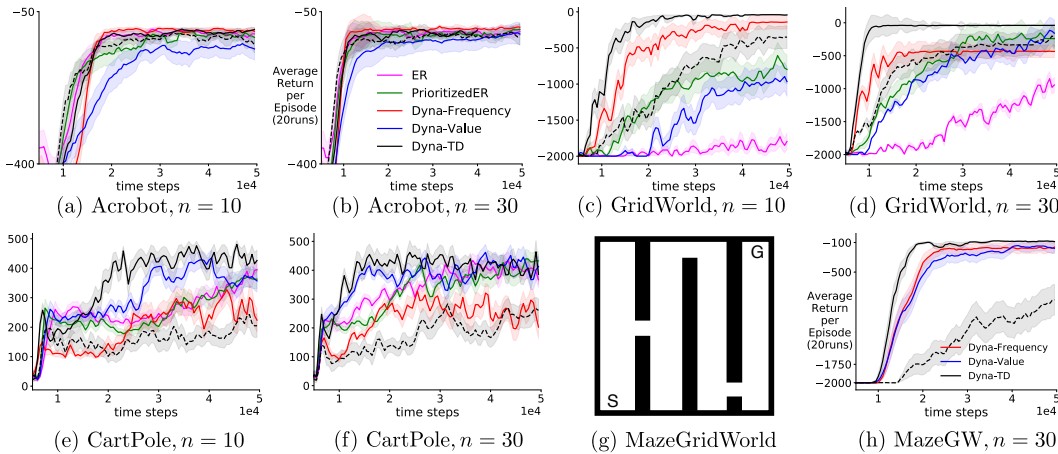

Figure 5: Evaluation learning curves on benchmark domains with planning updates $n = 10, 30$. The **dashed** line denotes Dyna-TD with an online learned model. All results are averaged over 20 random seeds. Figure(g) shows MazeGridWorld(GW) taken from Pan et al. (2020) and the learning curves are in (h). On MazeGW, we do not show model-free baselines as it is reported that model-free baselines do significantly worse than Dyna variants (Pan et al., 2020). We do reproduce the result of Dyna-Frequency from that paper.

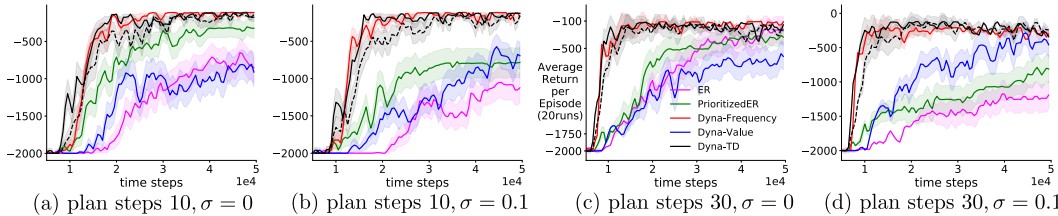

Figure 6: Evaluation learning curves on Mountain Car with different number of planning updates and different reward noise variance. At each time step, the reward is sampled from the Gaussian $N(-1, \sigma)$. $\sigma = 0$ indicates deterministic reward. All results are averaged over 20 random seeds.

Our interpretations of those observations are as follows. First, we can think about the case where some states have high value but low TD error. Dyna-Value may still frequently generate those states; this can waste samples and even incur sampling distribution bias, which can lead to a sub-optimal policy. This sub-optimality can be clearly observed on Acrobot, GridWorld and MazeGridWorld. Similar reasoning applies to Dyna-Frequency. Second, for Dyna-Frequency, as indicated by the original paper Pan et al. (2020), the gradient or Hessian norm have very different numerical scales and highly depend on the choice of the function approximator or domain. This indicates that the algorithm requires finely tuned parameter settings, as the testing domain is varied, which possibly explains its inconsistent performances across domains. Third, notice that even though each algorithm runs the same number of planning steps, the model-based algorithms perform significantly better. This provides evidence for the benefits of leveraging the generalization power of the learned value function and a model. In contrast, model-free methods can only utilize visited states.

**Robustness to Noise.** As a corresponding experiment to the supervised learning setting in Section 3, we show that our algorithm is more robust to increased noise variance than PrioritizedER. Figure 6 shows the evaluation learning curves on Mountain Car with planning steps $10, 30$ and reward noise standard deviation $\sigma \in \{0, 0.1\}$. We would like to identify three key observations. First, our algorithm's relative performance to PrioritizedER resembles the Full-PrioritizedL2 to PrioritizedL2 from the supervised learning setting, as Full-PrioritizedL2 is more robust to target noise than PrioritizedL2. Second, our algorithm achieves almost the same performance as Dyna-Frequency which is claimed to be robust to noise by Pan et al. (2020). Last, as observed on other environments, usually all algorithms can benefit from the increased number of planning steps; however, PrioritizedER and ER clearly degrade when using more planning steps with noise present.

**Practical Utility in Autonomous Driving Application.** We study the practical utility of our method in an autonomous driving application (Leurent, 2018) with an online learned model. As shown in Figure 7 (a), we test on the roundabout-v0 domain, where the agent (i.e. the green car) should learn to go through a roundabout without collisions while maintaining as high speed as possible. We would like to emphasize that there is a significantly lower number of car crashes with the policy learned by our algorithm on both domains as we show in Figure 7(b). This coincides with our intuition. The crash should incur high temporal difference error and our method of actively searching such states by gradient ascent. This ensures the agent gets sufficient training in this states during the planning stage, so that it learns to avoid them.

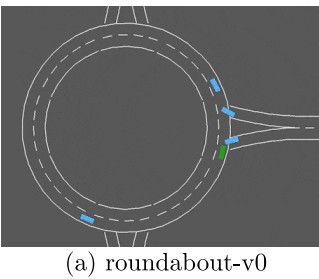

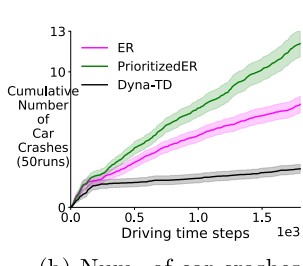

Figure 7: (a) shows the roundabout domain, where $\mathcal{S} \subset \mathbb{R}^{90}$. (b) shows the corresponding evaluation learning curves in terms of number of car crashes as a function of driving time steps. The results are averaged over 50 random seeds. The shade indicates standard error.

(a) roundabout-v0  (b) Num. of car crashes

## 6  DISCUSSION

In this work, we provide theoretical justification for why prioritized ER can help improve sample efficiency. We identify crucial factors for it to be effective: sample space coverage and thorough priority updating. We then propose to sample states by Langevin dynamics and conduct experiments to show our method's efficacy. There are several interesting directions for future work. One is to study the effects of model error on sample efficiency with our search control strategy. Another is to apply our method with a feature-to-feature model, which can improve our method's scalability. On the theory side, our cubic objective explains the original TD-error based prioritized ER (Schaul et al., 2016). However, there are other types of choices beyond TD-error based prioritization, such as distribution location or reward-based prioritization (Lambert et al., 2020). Whether these alternative prioritizations can also be formulated as surrogate objectives are interesting future directions. In a concurrent work, Fujimoto et al. (2020) established an equivalence between loss functions and sampling distributions, which bears similarities to our Theorem 1. However, it is not clear if similar optimization benefits shown in our Theorem 2 are enjoyed by more general loss functions and sampling distributions, which requires further investigations.

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

## A  Appendix

In Section A.1, we introduce some background in Dyna architecture. We briefly discuss Langevin dynamics and its computation cost in our case in Section A.2. We then provide the full proof of Theorem 2 in Section A.4. We present the proof for Theorem 4 in Section A.5. Details for reproducible research are in Section A.6. We provide supplementary experimental results in Section A.7.

### A.1  Background in Dyna

Dyna integrates model-free and model-based policy updates in an online RL setting (Sutton, 1990). As shown in Algorithm 2, at each time step, a Dyna agent uses the real experience to learn a model and performs model-free policy update. During the *planning* stage, simulated experiences are acquired from the model to further improve the policy. It should be noted that the concept of planning refers to any computational process which leverages a model to improve policy, according to Sutton & Barto (2018). The mechanism of generating states or state-action pairs from which to query the model is called *search-control*, which is of critical importance to the sample efficiency. There are abundant existing works (Moore & Atkeson, 1993; Sutton et al., 2008; Gu et al., 2016; Pan et al., 2018; Corneil et al., 2018; Goyal et al., 2019; Janner et al., 2019; Pan et al., 2019) report different level of sample efficiency improvements by using different way of generating hypothetical experiences during the planning stage.

---

**Algorithm 2** Tabular Dyna

Initialize $Q(s, a)$; initialize model $\mathcal{M}(s, a), \forall (s, a) \in \mathcal{S} \times \mathcal{A}$
**while** true **do**
 observe $s$, take action $a$ by $\epsilon$-greedy w.r.t $Q(s, \cdot)$
 execute $a$, observe reward $R$ and next State $s'$
 Q-learning update for $Q(s, a)$
 update model $\mathcal{M}(s, a)$ (i.e. by counting)
 store $(s, a)$ into search-control queue
 **for** i=1:d **do**
  sample $(\tilde{s}, \tilde{a})$ from search-control queue
  $(\tilde{s}', \tilde{R}) \leftarrow \mathcal{M}(\tilde{s}, \tilde{a})$ // simulated transition
  Q-learning update for $Q(\tilde{s}, \tilde{a})$ // planning update

---

### A.2  Discussion on the Langevin Dynamics

Define a SDE: $\mathrm{d}W(t) = \nabla U(W_t)\mathrm{d}t + \sqrt{2}\mathrm{d}B_t$, where $B_t \in \mathbb{R}^d$ is a $d$-dimensional Brownian motion and $U$ is a continuous differentiable function. It turns out that the Langevin diffusion $(W_t)_{t \geq 0}$ converges to a unique invariant distribution $p(x) \propto \exp(U(x))$ (Chiang et al., 1987). By applying the Euler-Maruyama discretization scheme to the SDE, we acquire the discretized version $Y_{k+1} = Y_k + \alpha_{k+1}\nabla U(Y_k) + \sqrt{2\alpha_{k+1}}Z_{k+1}$ where $(Z_k)_{k \geq 1}$ is an i.i.d. sequence of standard $d$-dimensional Gaussian random vectors and $(\alpha_k)_{k \geq 1}$ is a sequence of step sizes. It has been proved that the limiting distribution of the sequence $(Y_k)_{k \geq 1}$ converges to the invariant distribution of the underlying SDE Roberts (1996); Durmus & Moulines (2017). As a result, considering $U(\cdot)$ as $\delta(\cdot)$, $Y$ as $s$ completes the proof for Theorem 3.

**Computational time cost.** It should be noted that the Langevin Dynamics Monte Carlo method we used for generating states does introduce additional computation time cost. However, in the main body of the paper, we already show that the time cost worths 3(d).

In theory, the computational time is reasonably small. Although each gradient ascent step takes one backpropagation, this backpropagation is w.r.t. a single state, not a mini-batch. Let the mini-batch size of updating DQN be $b$, and the number of gradient steps be $k$. If we assume one mini-batch update takes $\mathcal{O}(c)$, then the time cost of Dyna-TD is $\mathcal{O}(kc/b)$. We would like to highlight that, though our approach incur higher computational cost, but it is able use fewer samples (i.e. fewer physical interactions with real environment) to achieve better performance. The cost of physical interactions could be higher in practical situations. When really needed, there are intuitive engineering tricks to reduce the computational cost. For example, we can learn a low dimensional embedding first and

build a model in such space; and then the gradient ascent can be done w.r.t. the feature instead of the raw input/observation itself. To further save time, we can also sacrifice a bit the accuracy and do the gradient ascent every certain number of time steps to lower down the amortized cost. Reducing time cost is not our current focus.

### A.3 PROOF FOR THEOREM 1

**Theorem 1.** For a constant $c$ determined by $\theta, \mathcal{T}$, we have

$$\mathbb{E}_{(x,y)\sim uniform(\mathcal{T})}[\nabla_\theta(1/3)|f_\theta(x) - y|^3] = c\mathbb{E}_{(x,y)\sim q(x,y;\theta)}[\nabla_\theta(1/2)(f_\theta(x) - y)^2]$$

*Proof.* The proof is very intuitive. The expected gradient of the uniform sampling method is

$$\mathbb{E}_{(x,y)\sim uniform(\mathcal{T})}[\nabla_\theta(1/3)|f_\theta(x) - y|^3] = \frac{1}{n}\sum_{i=1}^n |f_\theta(x_i) - y_i|\nabla_\theta(f_\theta(x_i) - y_i)^2$$

$$\mathbb{E}_{(x,y)\sim q(x,y;\theta)}[\nabla_\theta(1/2)(f_\theta(x) - y)^2] = \sum_{i=1}^n q(x_i, y_i; \theta)\nabla_\theta(f_\theta(x_i) - y_i)^2$$

$$= \frac{1}{\sum_{i=1}^n |f_\theta(x_i) - y_i|}\sum_{i=1}^n |f_\theta(x_i) - y_i|\nabla_\theta(f_\theta(x) - y)^2$$

$$= \frac{n}{\sum_{i=1}^n |f_\theta(x_i) - y_i|}\mathbb{E}_{(x,y)\sim uniform(\mathcal{T})}[\nabla_\theta(1/3)|f_\theta(x) - y|^3]$$

Setting $c = \frac{\sum_{i=1}^n |f_\theta(x_i)-y_i|}{n}$ completes the proof. $\square$

### A.4 PROOF FOR THEOREM 2

**Theorem 2.** Consider the following two objectives: $\ell_2(x, y) \stackrel{def}{=} \frac{1}{2}(x - y)^2$, and $\ell_3(x, y) \stackrel{def}{=} \frac{1}{3}|x - y|^3$. Denote $\delta_t \stackrel{def}{=} |x_t - y|$, and $\tilde{\delta}_t \stackrel{def}{=} |\tilde{x}_t - y|$. Define the functional gradient flow updates on these two objectives:

$$\frac{dx_t}{dt} = -\eta\frac{d\{\frac{1}{2}(x_t - y)^2\}}{dx_t}, \frac{d\tilde{x}_t}{dt} = -\eta\frac{d\{\frac{1}{3}|\tilde{x}_t - y|^3\}}{d\tilde{x}_t}. \tag{4}$$

Given error threshold $\epsilon \geq 0$, define the hitting time $t_\epsilon \stackrel{def}{=} \min_t\{t : \delta_t \leq \epsilon\}$ and $\tilde{t}_\epsilon \stackrel{def}{=} \min_t\{t : \tilde{\delta}_t \leq \epsilon\}$. For any initial function value $x_0$ s.t. $\delta_0 > 1$, $\exists \epsilon_0 \in (0, 1)$ such that $\forall \epsilon > \epsilon_0, t_\epsilon \geq \tilde{t}_\epsilon$.

*Proof.* For the gradient flow update on the $\ell_2$ objective, we have,

$$\frac{d\ell_2(x_t, y)}{dt} = \frac{d\ell_2(x_t, y)}{d\delta_t} \cdot \frac{d\delta_t}{dx_t} \cdot \frac{dx_t}{dt} \tag{5}$$

$$= \delta_t \cdot \text{sgn}(x_t - y) \cdot [-\eta \cdot (x_t - y)] \tag{6}$$

$$= \delta_t \cdot \text{sgn}(x_t - y) \cdot [-\eta \cdot \text{sgn}(x_t - y) \cdot \delta_t] \tag{7}$$

$$= -\eta \cdot \delta_t^2 = -2 \cdot \eta \cdot \ell_2(x_t, y). \tag{8}$$

which implies,

$$\frac{d\{\ln \ell_2(x_t, y)\}}{dt} = \frac{1}{\ell_2(x_t, y)} \cdot \frac{d\ell_2(x_t, y)}{dt} = -2 \cdot \eta. \tag{9}$$

Taking integral, we have,

$$\ln \ell_2(x_t, y) - \ln \ell_2(x_0, y) = -2 \cdot \eta \cdot t, \tag{10}$$

which is equivalent to (letting $\delta_t = \epsilon$),

$$t_\epsilon \stackrel{def}{=} \frac{1}{2\eta} \cdot \ln\left\{\frac{\ell_2(x_0, y)}{\ell_2(x_t, y)}\right\} = \frac{1}{\eta} \cdot \ln\left\{\frac{\delta_0}{\delta_t}\right\} = \frac{1}{\eta} \cdot \ln\left\{\frac{\delta_0}{\epsilon}\right\}. \tag{11}$$

On the other hand, for the gradient flow update on the $\ell_3$ objective, we have,

$$\frac{d\ell_3(\tilde{x}_t, y)}{dt} = \frac{d\ell_3(\tilde{x}_t, y)}{d\tilde{\delta}_t} \cdot \frac{d\tilde{\delta}_t}{d\tilde{x}_t} \cdot \frac{d\tilde{x}_t}{dt} \tag{12}$$

$$= \tilde{\delta}_t^2 \cdot \text{sgn}(\tilde{x}_t - y) \cdot \left[ -\eta \cdot \tilde{\delta}_t^2 \cdot \text{sgn}(\tilde{x}_t - y) \right] \tag{13}$$

$$= -\eta \cdot \tilde{\delta}_t^4 = -3^{\frac{4}{3}} \cdot \eta \cdot (\ell_3(\tilde{x}_t, y))^{\frac{4}{3}} , \tag{14}$$

which implies,

$$\frac{d\{(\ell_3(\tilde{x}_t, y))^{-\frac{1}{3}}\}}{dt} = -\frac{1}{3} \cdot (\ell_3(\tilde{x}_t, y))^{-\frac{4}{3}} \cdot \frac{d\ell_3(\tilde{x}_t, y)}{dt} = 3^{\frac{1}{3}} \cdot \eta. \tag{15}$$

Taking integral, we have,

$$(\ell_3(\tilde{x}_t, y))^{-\frac{1}{3}} - (\ell_3(\tilde{x}_0, y))^{-\frac{1}{3}} = 3^{\frac{1}{3}} \cdot \eta \cdot t, \tag{16}$$

which is equivalent to (letting $\tilde{\delta}_t = \epsilon$),

$$\tilde{t}_\epsilon \stackrel{\text{def}}{=} \frac{1}{3^{\frac{1}{3}} \cdot \eta} \cdot \left[ (\ell_3(\tilde{x}_t, y))^{-\frac{1}{3}} - (\ell_3(\tilde{x}_0, y))^{-\frac{1}{3}} \right] = \frac{1}{\eta} \cdot \left( \frac{1}{\tilde{\delta}_t} - \frac{1}{\delta_0} \right) = \frac{1}{\eta} \cdot \left( \frac{1}{\epsilon} - \frac{1}{\delta_0} \right). \tag{17}$$

Then we have,

$$t_\epsilon - \tilde{t}_\epsilon = \frac{1}{\eta} \cdot \ln \left\{ \frac{\delta_0}{\epsilon} \right\} - \frac{1}{\eta} \cdot \left( \frac{1}{\epsilon} - \frac{1}{\delta_0} \right) \tag{18}$$

$$= \frac{1}{\eta} \cdot \left[ \left( \ln \frac{1}{\epsilon} - \frac{1}{\epsilon} \right) - \left( \ln \frac{1}{\delta_0} - \frac{1}{\delta_0} \right) \right]. \tag{19}$$

Define the function $f(x) = \ln \frac{1}{x} - \frac{1}{x}, x > 0$ is continuous and $\max_{x>0} f(x) = f(1) = -1$. We have $\lim_{x \to 0} f(x) = \lim_{x \to \infty} f(x) = -\infty$, and $f(\cdot)$ is monotonically increasing for $x \in (0, 1]$ and monotonically decreasing for $x \in (1, \infty)$.

Given $\delta_0 > 1$, we have $f(\delta_0) < f(1) = -1$. Using the intermediate value theorem for $f(\cdot)$ on $(0, 1]$, we have $\exists \epsilon_0 < 1$, such that $f(\epsilon_0) = f(\delta_0)$. Since $f(\cdot)$ is monotonically increasing on $(0, 1]$ and monotonically decreasing on $(1, \infty)$, for any $\epsilon \in [\epsilon_0, \delta_0]$, we have $f(\epsilon) \geq f(\delta_0)$.[4] Hence we have,

$$t_\epsilon - \tilde{t}_\epsilon = \frac{1}{\eta} \cdot [f(\epsilon) - f(\delta_0)] \geq 0. \qquad \square$$

**Remark 1.** Figure 8 shows the function $f(x) = \ln \frac{1}{x} - \frac{1}{x}, x > 0$. Fix arbitrary $x' > 1$, there will be another root $\epsilon_0 < 1$ s.t. $f(\epsilon_0) = f(x')$. However, there is no real-valued solution for $\epsilon_0$. The solution in $\mathbb{C}$ is $\epsilon_0 = -\frac{1}{W(\log 1/\delta_0 - 1/\delta_0 - \pi i)}$, where $W(\cdot)$ is a Wright Omega function. Hence, finding the exact value of $\epsilon_0$ would require a definition of ordering on complex plane. Our current theorem statement is sufficient for the purpose of characterizing convergence rate. The theorem states that there always exists some desired low error level $< 1$, minimizing the square loss converges slower than the cubic loss.

## A.5 Proof for Theorem 4

We now provide the error bound for Theorem 4. We denote the transition probability distribution under policy $\pi$ with the true model as $\mathcal{P}^\pi(r, s'|s)$; denote that with the learned model as $\hat{\mathcal{P}}^\pi(r, s'|s)$. Let $p(s)$ and $\hat{p}(s)$ be the convergent distributions described in Theorem 3 by using true model and learned model respectively. Let $d_{tv}(\cdot, \cdot)$ be the total variation distance between two probability distributions. Define $u(s) \stackrel{\text{def}}{=} |\delta(s, y(s))|, \hat{u}(s) \stackrel{\text{def}}{=} |\delta(s, \hat{y}(s))|, Z \stackrel{\text{def}}{=} \int_{s \in \mathcal{S}} u(s)ds, \hat{Z} \stackrel{\text{def}}{=} \int_{s \in \mathcal{S}} \hat{u}(s)ds$. Then we have the following bound.

---

[4]Note that $\epsilon < \delta_0$ by the design of using gradient descent updating rule. If the two are equal, $t_\epsilon = \tilde{t}_\epsilon = 0$ holds trivially.

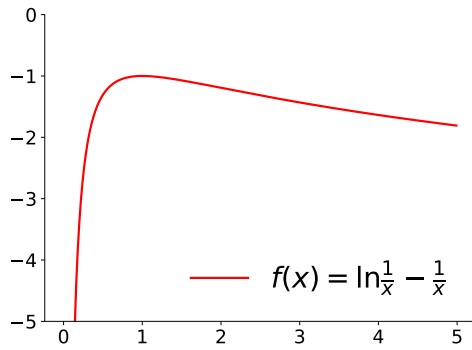

Figure 8: The function $f(x) = \ln \frac{1}{x} - \frac{1}{x}, x > 0$. The function reaches maximum at $x = 1$.

**Theorem 4.** Assume: 1) the reward magnitude is bounded $|r| \leq R_{max}$ and define $V_{max} \stackrel{\text{def}}{=} \frac{R_{max}}{1-\gamma}$; 2) the largest model error for a single state is some small value: $\epsilon_s \stackrel{\text{def}}{=} \max_s d_{tv}(\mathcal{P}^\pi(\cdot|s), \hat{\mathcal{P}}^\pi(\cdot|s))$ and the total model error is bounded, i.e. $\epsilon \stackrel{\text{def}}{=} \int_{s \in \mathcal{S}} \epsilon_s ds < \infty$. Then

$$\forall s \in \mathcal{S}, |p(s) - \hat{p}(s)| \leq \min(\frac{V_{max}(p(s)\epsilon + \epsilon_s)}{\hat{Z}}, \frac{V_{max}(\hat{p}(s)\epsilon + \epsilon_s)}{Z})$$

.

*Proof.* First, we bound the estimated temporal difference error. Fix an arbitrary state $s \in \mathcal{S}$, it is sufficient the consider the case $u(s) > \hat{u}(s)$, then

$$|u(s) - \hat{u}(s)| = u(s) - \hat{u}(s)$$
$$= \mathbb{E}_{(r,s') \sim \mathcal{P}^\pi}[r + \gamma v^\pi(s')] - \mathbb{E}_{(r,s') \sim \hat{\mathcal{P}}^\pi}[r + \gamma v^\pi(s')]$$
$$= \int_{s,r} (r + \gamma v^\pi(s))(\mathcal{P}^\pi(s', r|s) - \hat{\mathcal{P}}^\pi(s', r|s))ds'dr$$
$$\leq (R_{max} + \gamma \frac{R_{max}}{1-\gamma}) \int_{s,r} (\mathcal{P}^\pi(s', r|s) - \hat{\mathcal{P}}^\pi(s', r|s))ds'dr$$
$$\leq V_{max} d_{tv}(\mathcal{P}^\pi(\cdot|s), \hat{\mathcal{P}}^\pi(\cdot|s)) \leq V_{max}\epsilon_s$$

Now, we show that $|Z - \hat{Z}| \leq V_{max}\epsilon$.

$$|Z - \hat{Z}| = |\int_{s \in \mathcal{S}} u(s)ds - \int_{s \in \mathcal{S}} \hat{u}(s)ds| = |\int_{s \in \mathcal{S}} (u(s) - \hat{u}(s))ds|$$
$$\leq \int_{s \in \mathcal{S}} |u(s) - \hat{u}(s)|ds \leq V_{max} \int_{s \in \mathcal{S}} \epsilon_s ds = V_{max}\epsilon$$

Consider the case $p(s) > \hat{p}(s)$ first.

$$p(s) - \hat{p}(s) = \frac{u(s)}{Z} - \frac{\hat{u}(s)}{\hat{Z}}$$
$$\leq \frac{u(s)}{Z} - \frac{u(s) - V_{max}\epsilon_s}{\hat{Z}} = \frac{u(s)\hat{Z} - u(s)Z + ZV_{max}\epsilon_s}{Z\hat{Z}}$$
$$\leq \frac{u(s)V_{max}\epsilon + ZV_{max}\epsilon_s}{Z\hat{Z}} = \frac{V_{max}(p(s)\epsilon + \epsilon_s)}{\hat{Z}}$$

Meanwhile, below inequality should also hold:

$$p(s) - \hat{p}(s) = \frac{u(s)}{Z} - \frac{\hat{u}(s)}{\hat{Z}} \leq \frac{\hat{u}(s) + V_{max}\epsilon_s}{Z} - \frac{\hat{u}(s)}{\hat{Z}}$$

$$= \frac{\hat{u}(s)\hat{Z} - \hat{u}(s)Z + \hat{Z}V_{max}\epsilon_s}{Z\hat{Z}} \leq \frac{V_{max}(\hat{p}(s)\epsilon + \epsilon_s)}{Z}$$

Because both the two inequalities must hold, when $p(s) - \hat{p}(s) > 0$, we have:

$$p(s) - \hat{p}(s) \leq \min(\frac{V_{max}(p(s)\epsilon + \epsilon_s)}{\hat{Z}}, \frac{V_{max}(\hat{p}(s)\epsilon + \epsilon_s)}{Z})$$

It turns out that the bound is the same when $p(s) \leq \hat{p}(s)$. This completes the proof. $\square$

**Remark.** This bound actually indicates that $|p(s) - \hat{p}(s)|$ should be small. Because if $p(s)$ is much larger $\hat{p}$, then we may expect the second term in the min function would be chosen. During early learning, although the model error can be large, but $\hat{Z}, Z$ should be also very large. The total model error is scaled by $p(s)$ or $\hat{p}(s)$ and it should be small. We may expect a nice approximation even when the model is not that perfectly learned.

## A.6    REPRODUCIBLE RESEARCH

Our implementations are based on tensorflow with version 1.13.0 (Abadi et al., 2015). We use Adam optimizer (Kingma & Ba, 2014) for all experiments.

### A.6.1    REPRODUCE EXPERIMENTS BEFORE SECTION 5

**Supervised learning experiment.**    For the supervised learning experiment shown in section 3, we use $32 \times 32$ tanh units neural network, with learning rate swept from $\{0.01, 0.001, 0.0001, 0.00001\}$ for all algorithms. We compute the constant $c$ as specified in the Theorem 1 at each time step for Cubic loss. We compute the testing error every 500 iterations/mini-batch updates and our evaluation learning curves are plotted by averaging 50 random seeds. For each random seed, we randomly split the dataset to testing set and training set and the testing set has 1k data points. Note that the testing set is not noise-contaminated.

**Reinforcement Learning experiments in Section 3.**    We use a particularly small neural network $16 \times 16$ to highlight the issue of incomplete priority updating. Intuitively, a large neural network may be able to memorize each state's value and thus updating one state's value is less likely to affect others. We choose a small neural network, in which case a complete priority updating for all states should be very important. We set the maximum ER buffer size as 10k and mini-batch size as 32. The learning rate is 0.001 and the target network is updated every 1k steps.

**Distribution distance computation in Section 4.**    We now introduce the implementation details for Figure 3. The distance is estimated by the following steps. First, in order to compute the desired sampling distribution, we discretize the domain into $50 \times 50$ grids and calculate the absolute TD error of each grid (represented by the left bottom vertex coordinates) by using the true environment model and the current learned $Q$ function. We then normalize these priorities to get probability distribution $p^*$. Note that this distribution is considered as the desired one since we have access to all states across the state space with priorities computed by current Q-function at each time step. Second, we estimate our sampling distribution by randomly sampling 3k states from search-control queue and count the number of states falling into each discretized grid and normalize these counts to get $p_1$. Third, for comparison, we estimate the sampling distribution of the conventional prioritized ER (Schaul et al., 2016) by sampling 3k states from the prioritized ER buffer and count the states falling into each grid and compute its corresponding distribution $p_2$ by normalizing the counts. Then we compute the distances of $p_1, p_2$ to $p^*$ by two weighting schemes: 1) on-policy weighting: $\sum_{j=1}^{2500} d^\pi(s_j)|p_i(s_j) - p^*(s_j)|, i \in \{1, 2\}$, where $d^\pi$ is approximated by uniformly sample 3k states from a recency buffer and normalizing their visitation counts on the discretized GridWorld; 2) uniform weighting: $\frac{1}{2500} \sum_{j=1}^{2500} |p_i(s_j) - p^*(s_j)|, i \in \{1, 2\}$. We examine the two weighting schemes because of two considerations: for the on-policy weighting, we concern about the asymptotic convergent

behavior and want to down-weight those states with relatively high TD error but get rarely visited as the policy gets close to optimal; uniform weighting makes more sense during early learning stage, where we consider all states are equally important and want the agents to sufficiently explore the whole state space.

### A.6.2 Reproduce experiments in Section 5

For our algorithm, the pseudo-code with concrete parameter settings is presented in Algorithm 4.

**Common settings.** For all domains other than roundabout-v0, we use $32 \times 32$ neural network with ReLu hidden units except the Dyna-Frequency which uses tanh units as suggested by the author (Pan et al., 2020).[5] Except the output layer parameters which were initialized from a uniform distribution $[-0.003, 0.003]$, all other parameters are initialized using Xavier initialization (Glorot & Bengio, 2010). We use mini-batch size $b = 32$ and maximum ER buffer size 50k. All algorithms use target network moving frequency 1000 and we sweep learning rate from $\{0.001, 0.0001\}$. We use warm up steps $= 5000$ (i.e. random action is taken in the first 5k time steps) to populate the ER buffer before learning starts. We keep exploration noise as 0.1 without decaying.

**Meta-parameter settings.** For our algorithm Dyna-TD, we are able to keep the same parameter setting across all benchmark domains: $\alpha_a = 0.1, c = 20$ and learning rate 0.001. For all Dyna variants, we fetch the same number of states ($m = 20$) from hill climbing (i.e. search-control process) as Dyna-TD does, and use $\epsilon_a = 0.1$ and set the maximum number of gradient step as $k = 100$ unless otherwise specified.

Our Prioritized ER is implemented as the proportional version with sum tree data structure. To ensure fair comparison, since all model-based methods are using mixed mini-batch of samples, we use prioritized ER without importance ratio but half of mini-batch samples are uniformly sampled from the ER buffer as a strategy for bias correction. For Dyna-Value and Dyna-Frequency, we use the setting as described by the original papers.

For the purpose of learning an environment model, we use a $64 \times 64$ ReLu units neural network to predict $s' - s$ and reward given a state-action pair $s, a$; and we use mini-batch size 128 and learning rate 0.0001 to minimize the mean squared error objective for training the environment model.

**Environment-specific settings.** All of the environments are from OpenAI (Brockman et al., 2016) except that: 1) the GridWorld envirnoment is taken from Pan et al. (2019) and the MazeGridWorld is from Pan et al. (2020); 2) Roundabout-v0 is from Leurent et al. (2019). For all OpenAI environments, we use the default setting except on Mountain Car where we set the episodic length limit to 2k. The GridWorld has state space $\mathcal{S} = [0, 1]^2$ and each episode starts from the left bottom and the goal area is at the top right $[0.95, 1.0]^2$. There is a wall in the middle with a hole to allow the agent to pass. MazeGridWorld is a more complicated version where the state and action spaces are the same as GridWorld, but there are two walls in the middle and it takes a long time for model-free methods to be successful. On the this domain, we use the same setting as the original paper for all Dyna variants. We use exactly the same setting as described above except that we change the $Q-$ network size to $64 \times 64$ ReLu units, and number of search-control samples is $m = 50$ as used by the original paper. We refer readers to the original paper (Pan et al., 2020) for more details.

On roundabout-v0 domain, we use $64 \times 64$ ReLu units for all algorithms and set mini-batch size as 64. For Dyna-TD, we start using the model after 5k steps and set $m = 100, \alpha_a = 1.0, k = 500$ and we do search-control every 10 environment time steps to reduce computational cost. To alleviate the effect of model error, we use only 16 out of 64 samples from the search-control queue in a mini-batch.

### A.7 Additional Experiments

The main purpose of the additional experiments here is to strengthen our claims from Section 3.

**Learning curve in terms of training error corresponding to Figure 2.** In Section 3.1, we show the learning curve in terms of testing error. We now show training error to closely match our theoretical result 1. As a supplement, we include the learning curve in terms of testing error in Figure 9.

---

[5]Note that this is one of its disadvantages: the search-control of Dyna-Frequency requires the computation of Hessian-gradient product and it is empirically observed that the Hessian is frequently zero when using ReLu as hidden units (Pan et al., 2020).

---

**Algorithm 3** Dyna-TD

---

**Input:** $m$: number of states to fetch through search-control; $B_{sc}$: empty search-control queue; $B_{er}$: ER buffer; $\epsilon_a$: threshold for accepting a state; initialize $Q$-network $Q_\theta$
**for** $t = 1, 2, \ldots$ **do**
    Observe $(s_t, a_t, s_{t+1}, r_{t+1})$ and add it to $B_{er}$
    // Hill climbing on absolute TD error
    Sample $s$ from $B_{er}$, $c \leftarrow 0, \tilde{s} \leftarrow s$
    **while** $c < m$ **do**
        $\hat{y} \leftarrow \mathbb{E}_{s', r \sim \hat{\mathcal{P}}(\cdot|s,a)}[r + \gamma \max_a Q_\theta(s', a)]$
        Update $s$ by rule equation 3
        **if** $s$ is out of the state space **then**
            Sample $s$ from $B_{er}$, $\tilde{s} \leftarrow s$ // restart
            **continue**
        **if** $||\tilde{s} - s||_2/\sqrt{d} \geq \epsilon_a$ **then**
            // $d$ is the number of state variables, i.e. $\mathcal{S} \subset \mathbb{R}^d$
            Add $s$ into $B_{sc}$, $\tilde{s} \leftarrow s$, $c \leftarrow c + 1$
    //$n$ planning updates
    **for** $n$ times **do**
        Sample a mixed mini-batch with half samples from $B_{sc}$ and half from $B_{er}$
        Update $Q$-network parameters by using the mixed mini-batch

---

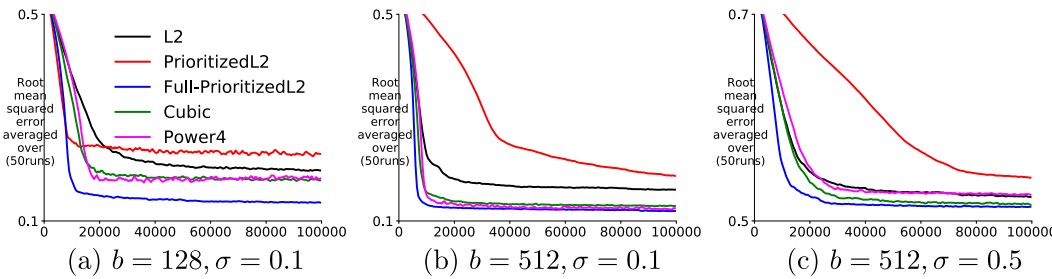

(a) $b = 128, \sigma = 0.1$      (b) $b = 512, \sigma = 0.1$      (c) $b = 512, \sigma = 0.5$

Figure 9: Figure(a)(b)(c) show the training RMSE as a function of number of mini-batch updates with different mini-batch sizes or Guassian noises with different $\sigma$ added to the training targets. The results are averaged over 50 random seeds. The standard error is small enough to get ignored.

**Learning curve with a larger neural network on the sin dataset.** We try to eliminate the effect of neural network size. Hence we use a larger neural network size ($128 \times 128$ tanh units) on the same sin dataset. As one can see from Figure 10, FullPrioritizedL2 still performs the best and when we increase the mini-batch size from 128 to 512, the high power objective versions still moves closer to FullPrioritizedL2, as we saw in Figure 2.

**Learning curve with on a real-world dataset.** To illustrate the generality of our Theorem 1, we also conduct tests on a frequently cited regression *Bike sharing dataset* Fanaee-T & Gama (2013). The data preprocessing is as follows. We remove attributes: date, index, year, weather situation 4, weekday 7, registered, casual. We use one-hot encoding for all categorical variables. We scale the target to $[0, 1]$ and scale it back when computing training errors. We use a $64 \times 64$ ReLu units neural network with mini-batch size 128 and learning rate 0.0001 for training.

It should be noted that the behavior of Cubic is consistent on the previous sin example and on this real world dataset: it gets closer to FullPrioritizedL2 as we increase the mini-batch size. Another observation is that the Power4 objective is highly variant on this domain, because the real world data should be noisy and the high order objective suffers. This observation corresponds to what we observed in Section 3.1, where we show that high power objective is sensitive to the noise.

---

**Algorithm 4** Dyna-TD with implementation details

---

**Input or notations:** $k = 20$: number search-control states to acquire by hill climbing, $k_b = 100$: the budget of maximum number of hill climbing steps; $\rho = 0.5$: percentage of samples from search-control queue, $d : \mathcal{S} \subset \mathbb{R}^d$; empty search-control queue $B_{sc}$ and ER buffer $B_{er}$

empirical covariance matrix: $\hat{\Sigma}_s \leftarrow \mathbf{I}$

$\mu_{ss} \leftarrow \mathbf{0} \in \mathbb{R}^{d \times d}, \mu_s \leftarrow \mathbf{0} \in \mathbb{R}^d$   (auxiliary variables for computing empirical covariance matrix, sample average will be maintained for $\mu_{ss}, \mu_s$)

$n_\tau \leftarrow 0$: count for parameter updating times, $\tau \leftarrow 1000$ target network updating frequency

$\epsilon_a \leftarrow 0$: threshold for accepting a state

Initialize $Q$ network $Q_\theta$ and target $Q$ network $Q_{\theta'}$

**for** $t = 1, 2, \ldots$ **do**

  Observe $(s, a, s', r)$ and add it to $B_{er}$

  $\mu_{ss} \leftarrow \frac{\mu_{ss}(t-1) + ss^\top}{t}, \mu_s \leftarrow \frac{\mu_s(t-1) + s}{t}$

  $\hat{\Sigma}_s \leftarrow \mu_{ss} - \mu_s \mu_s^\top$

  $\epsilon_a \leftarrow (1 - \beta)\epsilon_a + \beta ||s' - s||_2$ for $\beta = 0.001$

  // Hill climbing on absolute TD error

  Sample $s$ from $B_{er}, c \leftarrow 0, \tilde{s} \leftarrow s, i \leftarrow 0$

  **while** $c < k$ and $i < k_b$ **do**

    // since environment is deterministic, the environment model becomes a Dirac-delta distribution and we denote it as a deterministic function $\mathcal{M} : \mathcal{S} \times \mathcal{A} \mapsto \mathcal{S} \times \mathbb{R}$

    $s', r \leftarrow \mathcal{M}(s, a)$

    $\hat{y} \leftarrow r + \gamma \max_a Q_\theta(s', a)$

    // add a smooth constant $10^{-5}$ inside the logarithm

    $s \leftarrow s + \alpha_a \nabla_s \log(|\hat{y} - \max_a Q(s, a; \theta_t)| + 10^{-5}) + X, X \sim N(0, 0.01\hat{\Sigma}_s)$

    **if** $s$ is out of the state space **then**

      // restart hill climbing

      Sample $s$ from $B_{er}, \tilde{s} \leftarrow s$

      **continue**

    **if** $||\tilde{s} - s||_2 / \sqrt{d} \geq \epsilon_a$ **then**

      Add $s$ into $B_{sc}, \tilde{s} \leftarrow s, c \leftarrow c + 1$

    $i \leftarrow i + 1$

  **for** $n$ times **do**

    Sample a mixed mini-batch $b$, with proportion $\rho$ from $B_{sc}$ and $1 - \rho$ from $B_{er}$

    Update parameters $\theta$ (i.e. DQN update) with $b$

    $n_\tau \leftarrow n_\tau + 1$

    **if** $mod(n_\tau, \tau) == 0$ **then**

      $Q_{\theta'} \leftarrow Q_\theta$

---

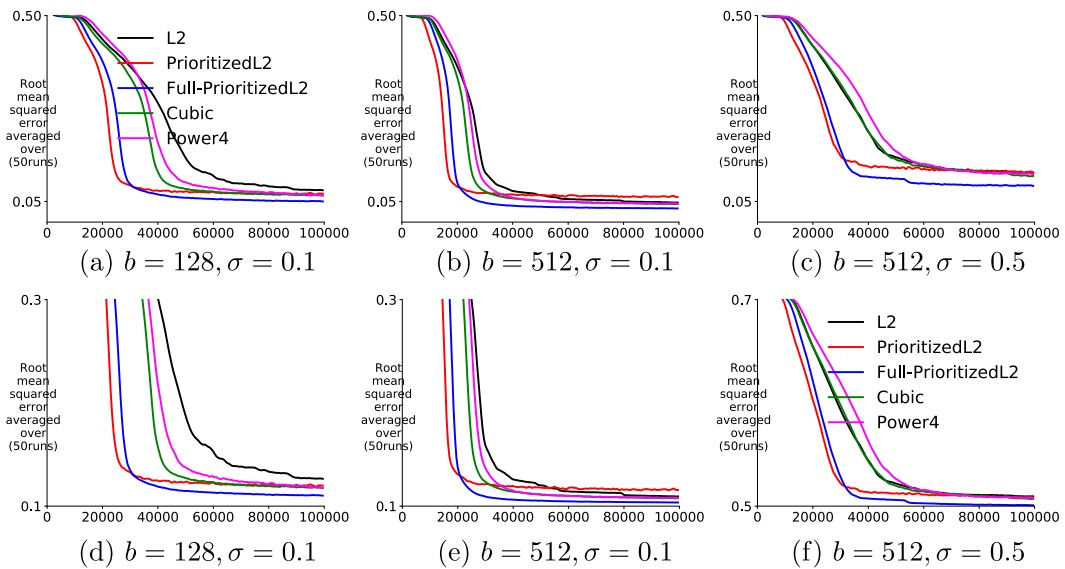

Figure 10: Figure(a)(b)(c) show the testing RMSE as a function of number of mini-batch updates with different mini-batch sizes or Guassian noises with different $\sigma$ added to the training targets. (d)(e)(f) show the training RMSE. The results are averaged over 50 random seeds. The standard error is small enough to get ignored. Note that the target variable in the testing set is not noise-contaminated.

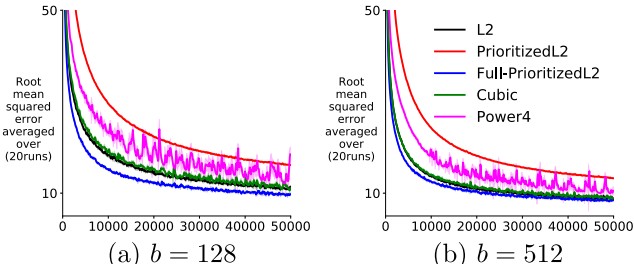

Figure 11: Figure(a)(b)(c) show the training RMSE as a function of number of mini-batch updates on the Bike sharing dataset. The results are averaged over 20 random seeds. The shade indicates standard error.

