# OpenReview forum: "Beyond Prioritized Replay: Sampling States in Model-Based RL via Simulated Priorities"
_ICLR.cc/2021/Conference — Reject_

### Official Review · AnonReviewer2 · 2020-10-18
**Is cubic objective good or bad?**

**Rating:** 5
**Confidence:** 3

**Review:**

Summary:

This paper investigates the search-control problem in Dyna-style reinforcement learning algorithms. They first provide a theoretical justification behind the error-based prioritization and propose a new sampling method based on gradient ascent of which optimization results are equivalent to samples drawn from the priority distribution. The suggested prioritization method is examined in various domains, namely, GridWorld, AcroBot, CartPole, MazeGridWorld, and roundabout-v0, and it shows a better sample efficiency in most domains.

Strength:

(+) The suggested method is based on the theoretical justification.

(+) The analysis of the out-dated priority is interesting and provides a better understanding of priority-based RL algorithms.

(+) The experimental results comparing the proposed Dyna-TD with other Dyna-variants corresponds to the theoretic justification they provided.

Concerns:

(-) While the authors provide a justification for the error-based prioritization in which the prioritization with l2-loss is equivalent to a cubic loss with uniform sampling, the motivation behind the error-based prioritization is rather weak. Specifically, in Theorem 2, it is stated that the cubic loss function provides fast early learning, but in Section 3.2, the opposite results are shown in which cubic or quadratic loss function is worse than Full-PrioritizedL2. What makes the specific form of error-based prioritization (absolute error normalized by the sum of the errors) best for learning?

(-) Related to the concern above, it is unclear how well Theorem 2 holds with parametric function approximation. The gap between Full-PrioritizedL2 and Cubic, which is theoretically equivalent, represents this problem. Where does the gap come from and why does the large batch size partially address this? At a glance, assuming that SGD finds the global optimum, the error should be the same for l2, l3, l4 when the optimization converges.

(-) It is a valid point that Dyna-TD addresses the insufficient sample space coverage problem of the ER methods and its variants via model M. However when the model has to be trained online, it is unclear whether the advantage still exists since the error of the model could hurt the hill-climbing process and the bootstrapped target value is more likely wrong. More thorough experiment results can alleviate this concern; how does the average return for the roundabout-v0 look like?; how well does other Dyna-variants algorithm work for the roundabout-v0 and MazeGW?; how about ER-based algorithms for MazeGW?; and most interestingly, how well does the algorithm work for environments having complex dynamics, such as atari-games? (less complex environments than the games would be also fine)

Typo:

(-) Missing parenthesis at the end of Theorem 3.

----------------
Post discussion comment:

I have decided not to update my score. While the theoretical analysis is interesting, I am not fully convinced about the utility of the proposed algorithm. It would be useful to have more experiments in the widely used benchmark such as atari or have better motivation explaining why faster convergence would help in the RL context as it would in supervised learning.

---

> ### Author Response · Authors · 2020-11-24
> **Response to R2**
>
> We appreciate your insightful and constructive feedback. We hope our response addresses the main weaknesses as you pointed out. We will address all minor issues, as you suggested.
>
> 1. "cubic vs. Full-PrioritizedL2"
>
> We would first point that Theorem 2 actually proves cubic achieves faster early learning than "uniform sampling L2" (not "Full-Prioritized L2"), which is consistent with Figure 2 (b)-(d), i.e., "Cubic" outperforms "L2".
>
> Theorem 1 shows cubic uniform sampling in expectation is equivalent with full prioritized L2, where "in expectation" requires batch-size to be large enough close to the full dataset size. We believe the reviewer mentioned in Fig 2(b) and (c), "Cubic" is worse than "Full-Prioritized L2". This is because the batch size is 128 or 512 (smaller than the full dataset size), as you can see that when we increase the mini-batch size, the two algorithms perform closer to each other. This is exactly as we expected.
>
> Based on the above results, we believe our theorems and experiments sufficiently support (i) "uniform sampling cubic" in expectation is equivalent with "full-prioritized L2"; (ii) "uniform sampling cubic" achieves faster early learning than "uniform sampling L2". We will make these points more clarified to avoid any confusion.
>
> 2. "Theorem 2 with parametric function approximation."
>
> As shown in our additional experiments above, the gap was from small batch-sizes and was closer by using a larger batch size on gradient cubic objective. We used parametric function (neural networks) in our experiments, which means that the evidence supports our arguments in practice.
>
> We agree with the reviewer that l2 and l3 preserve the same optimal solution. However, our point is their landscapes are different, which means the same algorithm will converge to some low error level with different speeds on those different objectives. And this is the high-level intuition why the "faster early learning" of the cubic is possible (and Theorem 2 rigorously formalized this intuition).
>
> 3. "model has to be trained online line."
>
> Our Dyna-TD with an online learned model does outperform those model-free baselines in many cases, as shown in Figure 4. However, we would like to emphasize that our work focuses on sampling distribution rather than learning a model. In fact, it is essential to separate the effect of model learning and the effect of the sampling distribution. By isolating other effects, we can clearly examine if a designed sampling distribution is effective or not. There are many model learning algorithms in MBRL literature. This should be considered as a separate question to study. Our work includes such results as supplementary.
>
> For roundabout-v0: the returns look similar for all algorithms. We can add this to the paper. However, we want to highlight Dyna-TD’s effect on reducing car crashes. The intuition is that, when there is a car crash, the TD error should be high, and our method would actively search for such situations.
>
> ER/PER in MazeGW: as the paper Frequency-based search control by Pan et al. indicates, the model-free variants are significantly worse than Dyna-Freq. And we show in our work that Dyna-TD is better than Dyna-Freq, and we reproduced the Dyna-Freq result.

---

### Official Review · AnonReviewer4 · 2020-10-26
**Incremental update to Dyna formulation that is hard to place in the literature.**

**Rating:** 4
**Confidence:** 3

**Review:**


Paper Summary:

This paper provides an update to a specific variant of model-based RL, Dyna-HC.
Dyna is a relatively old MBRL algorithm combining model-learning and model-free policy updates by leveraging the model as a simulator.
The paper shows how prioritized sampling from an experience replay buffer mirrors a cubic loss function and uses this insight to show prioritized state selection for simulated policy rollouts via hill climbing (gradient ascent).

-----

Review summary:
Incremental update to Dyna formulation in a busy paper that is hard to place in the literature.
Overall, the paper supports it's claims well, but it does not fit itself into the broader picture well. HC-Dyna is very recent work with only one small difference. The authors need to better motivate the work and draw clear boundaries to what was done in the past for it to be suitable for publication.

----
Comments:
1) The authors make some loose claims about model-based reinforcement learning. For example, the first sentence of the intro says MBRL "can significantly improve sample efficiency" but does not say what sample efficiency is being improved relative to. This is repeated in the last sentence of paragraph two of 2 Problem Formulation.

2) It should be said, this paper is well-written at a low-level. I found relatively few sentences confusing and few typos.

3) The results in Figure 5 do show a continued improvement of the Dynamics approach on more complex tasks. This is promising. It would be interesting to include other baselines in such a well-known task (such as cartpole). A reward of 500 in cartpole is strange, in my reading it generally maxes out at 200.
3b) why was the episodic limit of mountain car increased to 2k? From what is it increased?

4) In A.6.2 Reproduce - Common Settings, there is a lengthy list of training details. Were these required for performance?

5) in "regarding high power objectives" the authors say that the high objective requires a larger mini-batch, which is a weakness. Can this be partially mitigated by a bigger learning step in practice, because as the batch gets closer to the whole dataset, there is less noise and risk of taking bad gradient steps? Could this partially account for the difference in computation cost?

6) The algorithmic details in the paper should likely replace the algorithm provided "HC-dyna: generic framework," especially considering HC-dyna is very new and most readers are likely to be unfamiliar, I am not sure it gains much. Moving important content like this to the appendix is not desirable.

7) Figure 3d is very interesting and appreciated. Would like to see more RL papers do this. The numerical evaluation is relatively well done.

-----
Concerns:

1) This paper lacks context into the field and motivation for why I should care. This is highlighted by ending with a discussion section (which is mostly just a short summary and a future work section). Way conclusions should the reader draw?
1a) The title and introduction lead the reader to think there may be more comments on the broader area of MBRL. I understand that new experiments broadly are out of the scope of this paper with length considerations, but it would help the paper flow if there are intuitions for why this matters broadly.

2) This paper heavily relies on previous knowledge of Pan et. Al 2018 and 2019. Most of the content seems to be reviewing these papers, and I do not think the introduction did enough on differentiating them. This paper is well constructed, but these unclear boundaries make it difficult to accept for publication.

3) The section on PRIORITIZED SAMPLING AS A CUBIC OBJECTIVE does not seem particularly well motivated. Are there other types of prioritization than error-based prioritization. What of distribution location prioritization or reward-based prioritization as discussed in section 5 of this paper https://arxiv.org/pdf/2002.04523.pdf.
3a) Such error-based prioritization weights the learning on low-accuracy training points, but model accuracy does not guarantee reward.
3b) I do not find figure 1 insightful. This space could be better allocated to improve the flow, motivation, and conclusions of the paper.


-----
Minor Comments:
1) There are only a few typos
1a) first sentence of introduction "environment model" should either be hyphenated or environmental model
1b) "there is as yet" is wordy and seemingly lacking punctuation.
1c) "while keep the sampling distribution similar" at the end of section 4
2) in the empirical demonstration, paragraph 4, the authors say the learning rate is from the range {0.01,  0.001, 0.0001}, is this from a set? Is there any further optimization?
3) Figure 3c) is not mentioned in the caption.

----
Post discussion.
After reading some of the author responses I have decided not to update my score. I think the paper needs a bigger revision and more results to be above the acceptance threshold.

---

> ### Author Response · Authors · 2020-11-24
> **Response to R4**
>
> We appreciate your insightful and detailed feedback. We hope our response addresses your main concerns well. We will fix all minor issues in the final version of the paper.
>
> 1. motivation
>
> We thank the suggestion to present the impact of the work more directly. The following arguments will be added to show why researchers should care about our work.
>
> (a) Essentially, many RL algorithms are stochastic optimization algorithms. Sampling distribution (i.e., how to sample a mini-batch of samples to update parameters) affects the sample efficiency significantly. Note that we talk about sample efficiency as the number of real environment time steps v.s. Episodic return after training for this many environment time steps. Hence, high sample efficiency means that an agent can acquire high episodic return using a few real environment time steps/interactions.
>
> (b) Prioritized ER is highly cited; however, we did not see any work trying to understand why it can work to our best knowledge. We provided insight into prioritized ER and discovered its limitations. Then we propose to use Dyna-TD based on the Langevin dynamics Monte Carlo sampling method to overcome the limitations.
>
> 2. "differentiating the results of Pan et al. 2018 and 2019."
>
> Pan et al. 2018 and 2019 also used hill climbing on some learned functions to search for new states. The former does hill climbing on the learned value function, and the latter does hill climbing on the gradient/hessian norm of the learned value function. However, unlike our work, there is no connection between PER and the methods of Pan et al. 2018 and 2019. Our work explained why PER works (faster early learning than uniform sampling in Theorem 2) and its limitations (insufficient coverage and outdated priorities make the equivalence of Theorem 1 not hold). The motivation is different. Pan et al. 2018 and 2019 did not look into this problem.
>
> The proposed method is then also different from Pan et al. 2018 and 2019. Our method simulates prioritized ER by overcoming its limitations. Simultaneously, Pan et al. 2018 and 2019 are not necessarily related to ER, and their methods do not have a theoretical connection to convergence rate.
>
> 3. "other types of prioritization."
>
> We thank the reviewer for pointing out other prioritization types, which we would like to cite and explore further. However, we cannot investigate all those methods. The original PER paper [Schaul et al., 2016] is arguably the most cited but we obverse no theoretical understanding. As a result, we attempt to throw some insight, highlight its limitations, and propose promising solutions.
>
> Figure 1 shows that the point is in (b), cubic has a larger gradient magnitude than the squared objective when the error is large (or the iteration is far from the optimal solution). And this is the high-level reason why cubic can have a shorter hitting time than square (the gradient information is "stronger" for the cubic objective). We will make this point clearer.

---

### Official Review · AnonReviewer3 · 2020-10-28
**Interesting Investigation | Nice to have few more ablations**

**Rating:** 6
**Confidence:** 2

**Review:**


##########################################################################

Summary:

The paper proposes a new way of prioritization in experience replay and Dyna-style planning methods. In particular, it proposes to exploit a learned model to actively search for states with high expected errors. The states are then prioritized proportional to the expected errors. The authors motivate the approach by a theoretical/empirical observation that the prioritized optimization of L2 loss is equivalent to the direct optimization of cubic loss. More specifically they tackle (1) outdated priorities of training samples, (2) insufficient coverage of the sample space; which are identified as the main shortcomings of previously explored prioritization methods.

##########################################################################

Reasons for score:

Overall, I vote for borderline acceptance. I like the idea of exploiting a learned model to better prioritize the samples for planning. The experiments make a compelling case for a full update of the sampling priorities over partial updates and that Dyna-TD approximates a sample from the full distribution better. However, experiments fail to show how exactly insufficient coverage of the sample space is being tackled. Moreover, the empirical results in real domains fail to show a significant performance increase over Dyna-freq when a learned model is used. Assuming access to the real model for updating the priorities may not be fair to other approaches when the metric in question is the sample efficiency. Hopefully, the authors can address my concern in the rebuttal period.

##########################################################################

Pros:

1. The paper tackles one of the fundamental issues of training any RL agent - sampling mechanisms from the experience replays. For me, the problem itself is real and practical.
2. The proposed Dyna-TD prioritization scheme is novel such that it actively searches for states with a high expected error.  It is more compute efficient than updating priorities for all samples in an experience replay while maintaining a close approximation to the same. Moreover, it also provides a sampling method for sampling imaginary transitions from the model using the same method.
3. This paper provides comprehensive experiments, including theoretical analysis to show the effectiveness of the proposed framework. The proposed method outperforms previous prioritization methods with access to the real model and  is still comparable to the baselines with the learned online model;

##########################################################################

Cons:

1. Although the proposed method provides several ablation studies, I still suggest the authors conduct the following ablation studies to enhance the quality of the paper:

(a) How does Dyna-TD compare with Full update variants of Dyna-Value and Dyna-Freq prioritization mechanisms.

(b) Addition of Dyna-TD and Dyna-Freq methods for the Car Roundabout domain.

2. It is not clearly evident how insufficient coverage of sample space affects the learning process, and how Dyna-TD is specifically tackling this over Dyna-value or Dyna-Freq.

3. The use of a real model for prioritization is not fair when compared with baseline approaches that are not allowed to use the real model for search control.

##########################################################################

Questions during the rebuttal period:

Please address and clarify the cons above

---

> ### Author Response · Authors · 2020-11-24
> **Response to R3**
>
> Thank you for carefully reviewing our paper and providing valuable comments. We address your main concerns as follows.
>
> 1. "compare with full update variants of Dyna-Value and Dyna-Freq"
>
> We apologize for the insufficient description of the two existed works Dyna-Value and Dyna-Freq. The two variants do not have the outdated priorities issue as the model-free method PER. They are the same as Dyna-TD in the sense that they are also doing hill climbing for search-control based on the Q network parameter at the current time step. The difference is Dyna-TD is doing hill climbing on TD error magnitude, while Dyna-Value and Dyna-Freq are doing hill climbing on the value function and gradient/Hessian norm of value function respectively. As a result, the current empirical is already a fair comparison for these variants.
>
> 2. Addition of Dyna-Freq for Car roundabout domain.
>
> Dyna-Freq is not feasible in such a domain due to the large state space. Note that it involves computing 3rd order differentiation, which is very slow on 90-dimensional space.
>
> 3. "insufficient coverage of sample space"
>
> First, note that Dyna-TD is created to tackle the limitations of Prioritized ER, Dyna-Value and Dyna-Freq are just reasonable baselines to compare against.
>
> We agree with you that how exactly the insufficient coverage of sample space can affect the learning process is unclear and there does not exist a rigorous statement so far. This should be an open problem and deserve separate work. In our work, we empirically showed that the error-based prioritized method needs to have sufficient samples covering the whole sample space to show a clear advantage.
>
> 4. "the use of a real model is not fair”
>
> We indeed show our method Dyna-TD with an online learned model, in the black dashed line. It does outperform those model-free baselines in many cases. However, we would like to emphasize that our work focuses on sampling distribution, rather than how to learn a model. In fact, it is very important to separate the effect of model learning and the effect of the sampling distribution. By isolating model error effects, we can clearly examine if a designed sampling distribution is effective or not.

---

### Author Response · Authors · 2020-11-24
**Response to reviewers**

Dear Reviewers,

    We thank you for your time and effort to review our paper.

    The second discussion phase will end soon. Please let us know if you have any further concerns. We are happy to make further clarifications and strengthen our paper.

    We are aware that the time is tight. No matter if we have a chance to give a further response, we always appreciate your comments.

Thanks,
Authors

---

> ### Author Response · Authors · 2020-11-25
> **Response to reviewers: further update**
>
> Dear Reviewers,
>
> We thank you for your time and effort to review our paper.
>
> We noticed that on the ICLR website, it said: "During the rebuttal phase and for the camera-ready version, authors are allowed one additional page for the main text, for a strict upper limit of 9 pages."
>
> As a result, we updated our paper. We added more background about the previous works and discussed the differences on page 2. We also added figures to illuminate the broad sample coverage of our algorithm in Figure 4.
>
> For the discussion section, we added more discussions and references, as R4 suggested.
>
> Together with our rebuttal below, we should have addressed most of your concerns.
>
> Thanks,
> Authors

---

### Decision · Program_Chairs · 2021-01-07
**Final Decision**

**Decision:**

Reject

**Comment:**

# Quality:
While the paper presents an interesting approach, Reviewer 2 raised relevant questions about the assumption of the theoretical justification that needs to be thoroughly addressed.
Moreover, as noted by Reviewer4, the quality of the paper would also benefit from a more clear connection to existing model-based reinforcement learning literature, besides [Pan et al.]. For example, how much of the proposed approach and results can be applied in other algorithms?

# Clarity:
While the paper is generally well written and only minor suggestions from the reviewers should be implemented.

# Originality:
The proposed approach is a small but novel improvement over existing algorithms (to the best of the reviewers and my knowledge).

# Significance of this work:
The paper deal with a relevant and timely topic. However, it is currently very difficult to gauge the significance of this work, and it unclear if the results can be extended beyond toy benchmarks and to other RL algorithms. Several reviewers suggested additional experiments to strengthen the paper.

# Overall:
This paper deal with an interesting topic and presents new interesting results. However, the current manuscript is just below the acceptance threshold. Extending the experimental evaluation and improving the clarity of the paper would crucially increase the quality of the paper.